# Machine Learning by Two-Dimensional Hierarchical Tensor Networks: A Quantum Information Theoretic Perspective on Deep Architectures

## Abstract

The resemblance between the methods used in studying quantum-many body physics and in machine learning has drawn considerable attention. In particular, tensor networks (TNs) and deep learning architectures bear striking similarities to the extent that TNs can be used for machine learning. Previous results used one-dimensional TNs in image recognition, showing limited scalability and a request of high bond dimension. In this work, we train two-dimensional hierarchical TNs to solve image recognition problems, using a training algorithm derived from the multipartite entanglement renormalization ansatz (MERA). This approach overcomes scalability issues and implies novel mathematical connections among quantum many-body physics, quantum information theory, and machine learning. While keeping the TN unitary in the training phase, TN states can be defined, which optimally encodes each class of the images into a quantum many-body state. We study the quantum features of the TN states, including quantum entanglement and fidelity. We suggest these quantities could be novel properties that characterize the image classes, as well as the machine learning tasks. Our work could be further applied to identifying possible quantum properties of certain artificial intelligence methods.

## 1 Introduction

Over the past years, we have witnessed a booming progress in applying quantum theories and technologies to realistic problems. Paradigmatic examples include quantum simulators (Trabesinger et al., 2012) and quantum computers (Steane, 1998; Knill, 2010; Buluta et al., 2011) aimed at tackling challenging problems that are beyond the capability of classical digital computations. The power of these methods stems from the properties quantum many-body systems.

Tensor networks (TNs) belong to the most powerful numerical tools for studying quantum many-body systems (Verstraete et al., 2008; Orús, 2014a;b; Ran et al., 2017b). The main challenge lies in the exponential growth of the Hilbert space with the system size, making exact descriptions of such quantum states impossible even for systems as small as $\mathcal{O}(10^2)$ electrons. To break the "exponential wall", TNs were suggested as an efficient ansatz that lowers the computational cost to a polynomial dependence on the system size. Astonishing achievements have been made in studying, e.g. spins, bosons, fermions, anyons, gauge fields, and so on (Verstraete et al., 2008; Cirac & Verstraete, 2009; Orús, 2014b; Ran et al., 2017b) (Ran et al., 2017b). TNs are also exploited to predict interactions that are used to design quantum simulators (Ran et al., 2017a).

As TNs allowed the numerical treatment of difficult physical systems by providing layers of abstraction, deep learning achieved similar striking advances in automated feature extraction and pattern recognition (LeCun et al., 2015). The resemblance between the two approaches is beyond superficial. At theoretical level, there is a mapping between deep learning and the renormalization group (Bény, 2013), which in turn connects holography and deep learning (You et al., 2017; Gan & Shu, 2017), and also allows studying network design from the perspective of quantum entanglement (Levine et al., 2017). In turn, neural networks can represent quantum states (Carleo & Troyer, 2017; Chen et al., 2017; Huang & Moore, 2017; Glasser et al., 2017).

Most recently, TNs have been applied to solve machine learning problems such as dimensionality reduction (Cichocki et al., 2016; 2017), handwriting recognition (Stoudenmire & Schwab, 2016; Han et al., 2017). Through a feature mapping, an image described as classical information is transferred into a product state defined in a Hilbert space. Then these states are acted onto a TN, giving an output vector that determines the classification of the images into a predefined number of classes. Going further with this clue, it can be seen that when using a vector space for solving image recognition problems, one faces a similar "exponential wall" as in quantum many-body systems. For recognizing an object in the real world, there exist infinite possibilities since the shapes and colors change, in principle, continuously. An image or a gray-scale photo provides an approximation, where the total number of possibilities is lowered to $256^N$ per channel, with $N$ describing the number of pixels, and it is assumed to be fixed for simplicity. Similar to the applications in quantum physics, TNs show a promising way to lower such an exponentially large space to a polynomial one.

This work contributes in two aspects. Firstly, we derive an efficient quantum-inspired learning algorithm based on a hierarchical representation that is known as tree TN (TTN) (see, e.g., (Murg et al., 2015)). Compared with Refs. (Stoudenmire & Schwab, 2016; Han et al., 2017) where a one-dimensional (1D) TN (called matrix product state (MPS) (Östlund & Rommer, 1995)) is used, TTN suits more the two-dimensional (2D) nature of images. The algorithm is inspired by the multipartite entanglement renormalization ansatz (MERA) approach (Vidal, 2007; 2008; Cincio et al., 2008; Evenbly & Vidal, 2009), where the tensors in the TN are kept to be unitary during the training. We test the algorithm on both the MNIST (handwriting recognition with binary images) and CIFAR (recognition of color images) databases and obtain accuracies comparable to the performance of convolutional neural networks. More importantly, the TN states can then be defined that optimally encodes each class of images as a quantum many-body state, which is akin to the study of a duality between probabilistic graphical models and TNs (Robeva & Seigal, 2017). We contrast the bond dimension and model complexity, with results indicating that a growing bond dimension overfits the data. we study the representation in the different layers in the hierarchical TN with t-SNE (Van der Maaten & Hinton, 2008), and find that the level of abstraction changes the same way as in a deep convolutional neural network (Krizhevsky et al., 2012) or a deep belief network (Hinton et al., 2006), and the highest level of the hierarchy allows for a clear separation of the classes. Finally, we show that the fidelities between each two TN states from the two different image classes are low, and we calculate the entanglement entropy of each TN state, which gives an indication of the difficulty of each class.

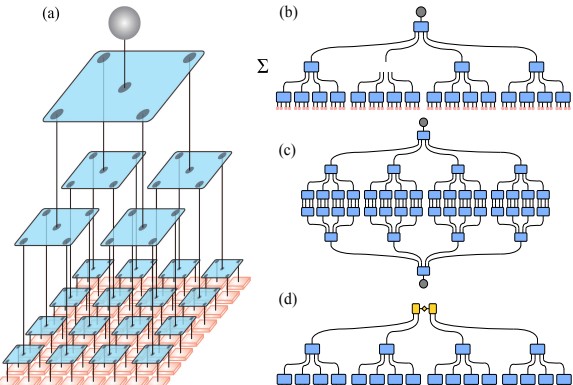

Figure 1: (Color online) The left figure (a) shows the configuration of TTN. The squares at the bottom represent the vectors obtained from the pixels of one image through the feature map. The sphere at the top represents the label. The right figures are (b) the illustrations of the environment tensor, (c) the schematic diagram of fidelity and (d) entanglement entropy calculation.

## 2 PRELIMINARIES OF TENSOR NETWORK AND MACHINE LEARNING

A TN is defined as a group of tensors whose indexes are shared and contracted in a specific way. TN can represent the partition function of a classical system, and also of a quantum many-body state which is mathematically a higher-dimensional vector. For the latter, one famous example is the

MPS that is written as $\Psi_{s_1 s_2 \cdots s_{N-1} s_N} = \sum_{\alpha_1 \cdots \alpha_{N-1}} A^{[1]}_{s_1 \alpha_1} A^{[2]}_{s_2 \alpha_1 \alpha_2} \cdots A^{[N-1]}_{s_{N-1} \alpha_{N-2} \alpha_{N-1}} A^{[N]}_{s_N \alpha_{N-1}}$. An MPS can simply be understood as a $d^N$-dimensional vector, with $d$ the dimension of $s_i$. Though the space increases exponentially with $N$, the cost of an MPS increases only polynomially as $N d D^2$ (with $D$ dimension of $\alpha_n$). When using it to describe an $N - site$ physical state, the un-contracted open indexes $\{s_n\}$ are called physical bonds that represent the physical Hilbert space[1], and contracted dummy indexes $\{\alpha_m\}$ are called virtual bonds that carry the quantum entanglement. MPS is essentially a 1D state representation. When applied to 2D systems, MPS suffers severe restrictions since one has to choose a snake-like 1D path that covers the 2D manifold. This issue is known in physics as the area law of entanglement entropy (Verstraete & Cirac, 2006; Hastings, 2007; Schuch et al., 2008).

A TTN (Fig. 1 (a)) provides a natural expression for 2D states, which we can write as a hierarchical structure of $K$ layers:

$$\Psi_{\alpha_{1,1} \cdots \alpha_{N_1, 4}} = \sum_{\{\alpha\}} \prod_{k=1}^{K} \prod_{n=1}^{N_k} T^{[k,n]}_{\alpha_{k+1,n'} \alpha_{k,n,1} \alpha_{k,n,2} \alpha_{k,n,3} \alpha_{k,n,4}}, \tag{1}$$

where $N_k$ is the number of tensors in the $k$-th layer.

To avoid the disaster brought by an extremely large number of indexes in a TN, we use the following symbolic and graphic conventions. A tensor is denoted by a bold letter without indexes, e.g., $\mathbf{T}$, whose elements are denoted by $T_{\alpha_1 \alpha_2 \ldots}$. Note a vector and a matrix are first- and second-order tensors with one and two indexes, respectively. When two tensors are multiplied together, the common indexes are to be contracted. One example is the inner product of two vectors, where $\sum_\alpha v^*_\alpha u_\alpha$ is simplified to $\mathbf{v}^\dagger \mathbf{u}$. We take the transpose of $\mathbf{v}$ because we always assume the vectors to be column vectors. Another example is the matrix product, where $X_{b_1 b_2} = \sum_\alpha M^{[1]}_{b_1 \alpha} M^{[2]}_{\alpha b_2}$ is simplified to $\mathbf{X} = \mathbf{M}^{[1]} \mathbf{M}^{[2]}$. $\alpha$ is an dummy index, and $b_1$ and $b_2$ are two open indexes. In the graphic representation, a tensor is a block connecting to several bonds. Each bond represents an index belonging to this tensor. The dummy indexes are represented by the shared bonds that connect to two different blocks. Following this convention, Eq. (1) can be simplified to $\mathbf{\Psi} = \prod_{k=1}^{K} \prod_{n=1}^{N_k} \mathbf{T}^{[k,n]}$.

Similar to the MPS, a TTN also provides a representation of a $d^N$-dimensional vector. The cost is also polynomial to $N$. One advantage is that the TTN bears a hierarchical structure and can be naturally built for 2D systems. In a TTN, each local tensor is chosen to have one upward index and four downward indexes. For representing a pure state, the tensor on the top only has four downward indexes. All the indexes except the downward ones of the tensors in the first layer are dummy and will be contracted. In our work, the TTN is slightly different from the pure state representation, by adding an upward index to the top tensor (Fig. 1 (a)). This added index corresponds to the labels in the supervised machine learning.

Before training, we need to prepare the data with a feature function that maps $N$ scalars ($N$ is the dimension of the images) to the tensor product of $N$ normalized vectors. The choice of the feature function is diversified: we chose the one used in Ref. (Stoudenmire & Schwab, 2016), where the dimension of each vector ($d$) can be controlled. Then, the space is transformed from that of $N$ scalars to a $d^N$-dimensional vector (Hilbert) space.

After "vectorizing" the $j$-th image in the dataset, the output for classification is a $\tilde{d}$-dimensional vector obtained by contracting this huge vector with the TTN, which reads as

$$\tilde{\mathbf{L}}^{[j]} = \mathbf{\Psi} \prod_{n=1}^{N} \mathbf{v}^{[j,n]}, \tag{2}$$

where $\{\mathbf{v}^{[j,n]}\}$ denotes the $n$-th vector given by the $j$-th sample. One can see that $\tilde{d}$ is the dimension of the upward index of the top tensor, and should equal to the number of the classes. We use the convention that the position of the maximum value gives the classification of the image predicted by the TTN, akin to a softmax layer in a deep learning network.

---

[1]Hilbert space is defined as the vector space spanned by the basis of quantum states. Each quantum state is described by a vector living in this space.

One choice of the cost function to be minimized is the square error, which is defined as

$$f = \sum_{j=1}^{J} |\tilde{\mathbf{L}}^{[j]} - \mathbf{L}^{[j]}|^2, \tag{3}$$

where $J$ is the number of training samples. $\mathbf{L}^{[j]}$ is a $\tilde{d}$-dimensional vector corresponding to the $j$-th label. For example, if the $j$-th sample belongs to the $p$-th class, $\mathbf{L}^{[j]}$ is defined as

$$L_\alpha^{[j]} = \begin{cases} 1, & \text{if } \alpha = p \\ 0, & \text{otherwise} \end{cases} \tag{4}$$

## 3 MERA-INSPIRED TRAINING ALGORITHM

Inspired by MERA (Vidal, 2007), we derive a highly efficient training algorithm. To proceed, let us rewrite the cost function in the following form

$$f = \sum_{j=1}^{J} \left( \prod_{nn'} \mathbf{v}^{[j,n']\dagger} \mathbf{\Psi}^\dagger \mathbf{\Psi} \mathbf{v}^{[j,n]} - 2 \prod_n \mathbf{L}^{[j]\dagger} \mathbf{\Psi} \mathbf{v}^{[j,n]} + 1 \right). \tag{5}$$

The third term comes from the normalization of $\mathbf{L}^{[j]}$, and we assume the second term is always real.

The dominant cost comes from the first term. We borrow the idea from the MERA approach to reduce this cost. Mathematically speaking, the central idea is to impose that $\mathbf{\Psi}$ is orthogonal, i.e., $\mathbf{\Psi}\mathbf{\Psi}^\dagger = \mathbf{I}$. Then $\mathbf{\Psi}$ is optimized with $\mathbf{\Psi}^\dagger\mathbf{\Psi} = \mathbf{I}$ satisfied in the valid subspace that optimizes the classification. By satisfying in the subspace, we do not require an identity from $\mathbf{\Psi}^\dagger\mathbf{\Psi}$, but mean $\sum_{j=1}^{J} \prod_{nn'} \mathbf{v}^{[j,n']\dagger} \mathbf{\Psi}^\dagger \mathbf{\Psi} \mathbf{v}^{[j,n]} \simeq \sum_{j=1}^{J} \prod_{nn'} \mathbf{v}^{[j,n']\dagger} \mathbf{v}^{[j,n]} = J$ under the training samples.

In MERA, a stronger constraint is used. With the TTN, each tensor has one upward and four downward indexes, which gives a non-square orthogonal matrix by grouping the downward indexes into a large one. Such tensors are called isometries and satisfy $\mathbf{T}\mathbf{T}^\dagger = \mathbf{I}$ after contracting all downwards indexes with its conjugate. When all the tensors are isometries, the TTN gives a unitary transformation that satisfies $\mathbf{\Psi}\mathbf{\Psi}^\dagger = \mathbf{I}$; it compresses a $d^N$-dimensional space to a $\tilde{d}$-dimensional one.

In this way, the first terms becomes a constant, and we only need to deal with the second term. The cost function becomes

$$f = -\sum_{j=1}^{J} \prod_n \mathbf{L}^{[j]\dagger} \mathbf{\Psi} \mathbf{v}^{[j,n]}. \tag{6}$$

Each term in $f$ is simply the contraction of one TN, which can be efficiently computed. We stress that independent of Eq. (3), Eq. (6) can be directly used as the cost function. This will lead to a more interesting picture connected to the condensed matter physics and quantum information theory.

From the physical point of view, the central idea of MERA is the renormalization group (RG) of the entanglement (Vidal, 2007). The RG flows are implemented by the isometries that satisfy $\mathbf{T}\mathbf{T}^\dagger = \mathbf{I}$. On one hand, the orthogonality makes the state remain normalized, a basic requirement of quantum states. On the other hand, the renormalization group flows can be considered as the compressions of the Hilbert space (from the downward to upward indexes). The orthogonality ensure that such compressions are unbiased with $\mathbf{T}^\dagger\mathbf{T} \simeq \mathbf{I}$ in the subspace. The difference from the identity characterizes the errors caused by the compressions. More discussions are given in Sec. 5.

The tensors in the TTN are updated alternatively to minimize Eq. (6). To update $\mathbf{T}^{[k,n]}$ for instance, we assume other tensors are fixed and define the *environment tensor* $\mathbf{E}^{[k,n]}$, which is calculated by contracting everything in Eq. (6) after taking out $\mathbf{T}^{[k,n]}$ (Fig. 1 (b)) (Evenbly & Vidal, 2009). Then the cost function becomes $f = -\text{Tr}(\mathbf{T}^{[k,n]}\mathbf{E}^{[k,n]})$. Under the constraint that $\mathbf{T}^{[k,n]}$ is an isometry, the solution of the optimal point is given by $\mathbf{T}^{[k,n]} = \mathbf{V}\mathbf{U}^\dagger$ where $\mathbf{V}$ and $\mathbf{U}$ are calculated from the singular value decomposition $\mathbf{E}^{[k,n]} = \mathbf{U}\mathbf{\Lambda}\mathbf{V}^\dagger$. At this point, we have $f = -\sum_a \Lambda_a$.

Then, the update of one tensor becomes the calculation of the environment tensor and its singular value decomposition. In the alternating process for updating all the tensors, some tricks are used

to accelerate the computations. The idea is to save some intermediate results to avoid repetitive calculations by taking advantage of the tree structure. Another important detail is to normalize the vector obtained each time by contracting four vectors with a tensor.

The strategy for building a multi-class classifier is the one-against-all classification scheme in machine learning. For each class, we train one TTN so that it recognizes whether an image belongs to this class. The output of Eq. (2) is a two-dimensional vector. We fix the label for a *yes* answer as $\mathbf{L}^{yes} = [1, 0]$. For $P$ classes, we will accordingly have $P$ TTNs, denoted by $\{\mathbf{\Psi}^{(p)}\}$. Then for recognizing an image (vectorized to $\{\mathbf{v}^{[n]}\}$), we define a $P$-dimensional vector $\mathbf{F}$ as

$$F_p = \mathbf{L}^{yes\dagger} \mathbf{\Psi}^{(p)} \prod_n \mathbf{v}^{[n]}. \tag{7}$$

The position of its maximal element gives which class the image belongs to.

---

**Algorithm 1** One-against-All

---

**Require:** $data$ : data points,
         $n$ : the number of data points
1: **for** $i = 0 \to 9$ **do**
2:     Train binary classifier $classifier_k$ corresponding to each handwritten digit
3: **end for**
4: **for** $j = 1 \to n$ **do**
5:     **for** $k = 0 \to 9$ **do**
6:        $output_k \leftarrow classifier_k(data_j)$;
7:     **end for**
8:     $label_j \leftarrow argmax(output(k))$;
9:     **return** $label_j$
10: **end for**

---

The scaling of both time complexity and space complexity is $O((b_v{}^5 + b_i^4 b_v)MN_T)$, where $M$ is the dimension of input vector; $b_v$ the dimension of virtual bond; $b_i$ the dimension of input bond; $N_T$ the number of training inputs.

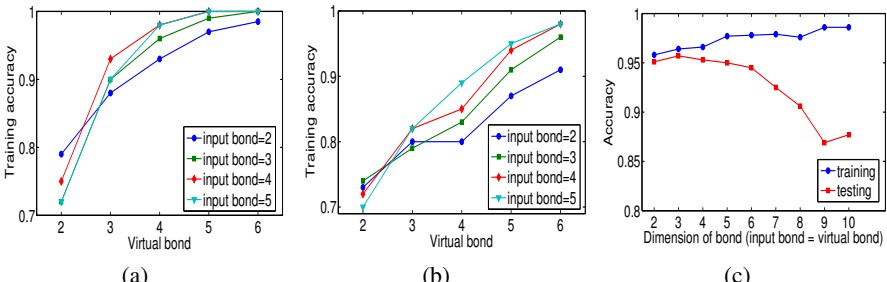

Figure 2: (a) Binary classification accuracy on CIFAR-10 with number of training samples=200; (b) Binary classification accuracy on CIFAR-10 with number of training samples=600; (c) Training and test accuracy as the function of the dimension of indexes on the MNIST dataset. The number of training samples is 1000 for each pair of classes.

## 4   Experiments on Image Recognition

Our approach to classify image data begins by mapping each pixel $x_j$ to a d-component vector $\phi^{s_j}(x_j)$. This feature map was introduced by (Stoudenmire & Schwab, 2016)) and defined as $\phi^{s_j}(x_j) = \sqrt{\binom{d-1}{s_j-1}}(cos(\frac{\pi}{2}x_j))^{d-s_j}(sin(\frac{\pi}{2}x_j))^{s_j-1}$, where $s_j$ runs from 1 to $d$. By using a larger $d$, the TTN has the potential to approximate a richer class of functions.

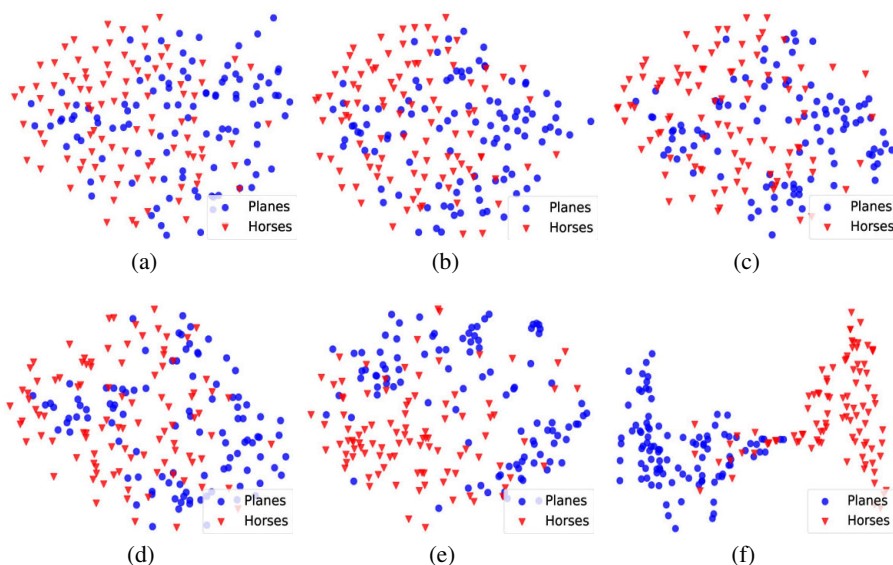

Figure 3: Embedding of data instances of CIFAR-10 by t-SNE corresponding to each layer in the TTN: (a) original data distribution and (b) the 1st, (c) 2nd, (d) 3rd, (e) 4th, and (f) 5th layer.

## 4.1 BENCHMARK ON CIFAR-10

To verify the representation power of TTNs, we used the CIFAR-10 dataset (Krizhevsky & Hinton, 2009). The dataset consists of 60,000 $32 \times 32$ RGB images in 10 classes, with 6,000 instances per class. There are 50,000 training images and 10,000 test images. Each RGB image was originally $32 \times 32$ pixels: we transformed them to grayscale. Working with gray-scale images reduced the complexity of training, with the trade-off being that less information was available for learning.

We built a TTN with five layers and used the MERA-like algorithm (Section 3) to train the model. Specifically, we built a binary classification model to investigate key machine learning and quantum features, instead of constructing a complex multiclass model. We found both the input bond (physical indexes) and the virtual bond (geometrical indexes) had a great impact on the representation power of TTNs, as showed in Fig. 2. This indicates that the limitation of representation power (learnability) of the TTNs is related to the input bond. The same way, the virtual bond determine how accurately the TTNs approximate this limitation.

From the perspective of tensor algebra, the representation power of TTNs depends on the tensor contracted from the entire TTN. Thus the limitation of this relies on the input bond. Furthermore, the TTNs could be considered as a decomposition of this complete contraction, and the virtual bond determine how well the TTNs approximate this. Moreover, this phenomenon could be interpreted from the perspective of quantum many-body theory: the higher entanglement in a quantum many-body system, the more representation power this quantum system has.

The sequence of convolutional and pooling layers in the feature extraction part of a deep learning network is known to arrive at higher and higher levels of abstractions that helps separating the classes in a discriminative learner (LeCun et al., 2015). This is often visualized by embedding the representation in two dimensions by t-SNE (Van der Maaten & Hinton, 2008), and by coloring the instances according to their classes. If the classes clearly separate in this embedding, the subsequent classifier will have an easy task performing classification at a high accuracy. We plotted this embedding for each layer in the TN in Fig. 3. We observe the same pattern as in deep learning, having a clear separation in the highest level of abstraction.

## 4.2 BENCHMARK ON MNIST

To test the generalization of TTNs on a benchmark dataset, we used the MNIST collection, which is widely used in handwritten digit recognition. The training set consists of 60,000 examples, and the

Table 1: 10-class classification on MNIST

| model | 0 | 1 | 2 | 3 | 4 | 5 | 6 | 7 | 8 | 9 | 10-class |
|---|---|---|---|---|---|---|---|---|---|---|---|
| Training accuracy (%) | 96 | 97 | 96 | 94 | 96 | 94 | 97 | 94 | 93 | 94 | 95 |
| Testing accuracy (%) | 97 | 97 | 95 | 93 | 95 | 95 | 96 | 94 | 93 | 93 | 92 |
| Input bond | 3 | 3 | 3 | 4 | 2 | 6 | 2 | 6 | 6 | 4 | / |
| Virtual bond | 3 | 3 | 4 | 4 | 3 | 6 | 3 | 6 | 6 | 6 | / |

test set of 10,000 examples. Each gray-scale image of MNIST was originally $28 \times 28$ pixels, and we rescaled them to $16 \times 16$ pixels for building TTNs with four layers on this scale. The MERA-like algorithm was used to train the model.

Similar to the last experiment, we built a binary model to show the performance of generalization. With the increase of bond dimension (both of the input bond and virtual bond), we found an apparent rise of training accuracy, which is consistent with the results in Fig. 2. At the same time, we observed the decline of testing accuracy. The increase of bond dimension leads to a sharp increase of the number of parameters and, as a result, it will give rise to overfitting and lower the performance of generalization. Therefore, one must pay attention to finding the optimal bond dimension – we can think of this as a hyperparameter controlling model complexity.

We choose the one-against-all strategy to build a 10-class model, which classify an input image by choosing the label for which the output is largest. Considering the efficiency and avoiding overfitting, we use the minimal values of $d$ (Table 1) to reach the training accuracy around $95\%$.

## 5 ENCODING IMAGES IN QUANTUM STATES: FIDELITY AND ENTANGLEMENT

Taking one trained TTN $\mathbf{\Psi}$ where the index for the labels is assumed to be $P$-dimensional, we can define $P$ normalized TTN vector (state) as

$$\mathbf{\Phi}^{[p]} = \mathbf{L}^{[p]\dagger} \mathbf{\Psi}. \tag{8}$$

In $\mathbf{\Phi}^{[p]}$, the upward index of the top tensor is contracted with the label ($\mathbf{L}^{[p]}$), giving a TN state that represents a normalized $d^N$-dimensional vector (pure quantum state).

The quantum state representations allow us to use quantum theories to study images and the related issues. Let us begin with the cost function. In Section 3, we started from a frequently used cost function in Eq. (3), and derived a cost function in Eq. (6). In the following, we show that Eq. (6) can be understood by the notion of fidelity. With Eq. (8), the cost function in Eq. (6) can be rewritten as $f = -\sum_j \mathbf{\Phi}^{[p]T} \prod_n \mathbf{v}^{[j,n]}$.

The fidelity between two states (normalized vectors) is defined as their inner product, thus each term in the summation is simply the fidelity (Steane, 1998; Bennett & DiVincenzo, 2000) between a vectorized image and the corresponding TTN state $\mathbf{\Phi}^{[p]}$. Considering that the fidelity measures the distance between two states, $\{\mathbf{\Phi}^{[p]}\}$ are the $P$ states that minimize the distance between each $\mathbf{\Phi}^{[p]}$ and the $p$-th vectorized images. In other words, the cost function is in fact the total fidelity, and $\mathbf{\Phi}^{[p]}$ is the quantum state (normalized vector) that optimally encodes the $p$-th class of images.

Note that due to the orthogonality, such $P$ states are orthogonal to each other, i.e., $\mathbf{\Phi}^{[p']\dagger} \mathbf{\Phi}^{[p]} = I_{p'p}$. This might trap us to a bad local minimum. For this reason, we propose the one-against-all strategy (see Algorithm 3). For each class, we have two TN states labeled *yes* and *no*, respectively, and in total $2P$ TN states. $\{\mathbf{\Phi}^{[p]}\}$ are then defined by taking the $P$ *yes*-labeled TN states. The elements of $\mathbf{F}$ in Eq. (7) are defined by the summation of the fidelity between $\mathbf{\Phi}^{[p]}$ and the class of vectorized images. In this scenario, the classification is decided by finding the $\mathbf{\Phi}^{[p]}$ that gives the maximal fidelity with the input image, while the orthogonal conditions among $\{\mathbf{\Phi}^{[p]}\}$ no longer exist.

Besides the algorithmic interpretation, fidelity may imply more intrinsic information. Without the orthogonality of $\{\mathbf{\Phi}^{[p]}\}$, the fidelity $\mathcal{F}_{p'p} = \mathbf{\Phi}^{[p']\dagger} \mathbf{\Phi}^{[p]}$ (Fig. 1 (c)) describes the differences between the quantum states that encode different classes of images. As shown in Fig. 4(a), $\mathcal{F}_{p'p}$ remains quite

small in most cases, indicating that the orthogonality still approximately holds. Still, some results are still relatively large, e.g., $\mathcal{F}_{4,9} = 0.1353$. We speculate this is closely related to the ways how the data are fed and processed in the TN. In our case, two image classes that have similar shapes will result in a larger fidelity, because the TTN essentially provides a *real-space renormalization flow*. In other words, the input vectors are still initially arranged and renormalized layer by layer according to their spatial locations in the image; each tensor renormalizes four nearest-neighboring vectors into one vector. Fidelity can be potentially applied to building a network, where the nodes are classes of images and the weights of the connections are given by the $\mathcal{F}_{p'p}$. This might provide a mathematical model on how different classes of images are associated to each other. We leave these questions for future investigations.

Another important concept of quantum mechanics is (bipartite) entanglement, a quantum version of correlations (Bennett & DiVincenzo, 2000). It is one of the key characters that distinguishes the quantum states from classical ones. Entanglement is usually given by a normalized positive-defined vector called entanglement spectrum (denoted as $\Lambda$), and is measured by the entanglement entropy $S = -\sum_a \Lambda_a^2 \ln \Lambda_a^2$. Having two subsystems, entanglement entropy measures the amount of information of one subsystem that can be gained by measuring the other subsystem. In the framework of TN, entanglement entropy determines the minimal dimensions of the dummy indexes needed for reaching a certain precision.

In our image recognition, entanglement entropy characterizes how much information of one part of the image we can gain by knowing the rest part of the image. In other words, if we only know a part of an image and want to predict the rest according to the trained TTN (the quantum state that encodes the corresponding class), the entanglement entropy measures how accurately this can be done. Here, an important analog is between knowing a part of the image and measuring the corresponding subsystem of the quantum state. Thus, the trained TTN might be used on image processing, e.g., to recover an image from a damaged or compressed lower-resolution version.

Fig. 4(b) shows the entanglement entropy for each class in the MNIST dataset. We computed two kinds of entanglement entropy marked by up-down and left-right. The first one denotes the entanglement between Upper part of the images with the lower part one. The later one denotes the entanglement between left part with the right part. With the TTN, the entanglement spectrum is simply the singular values of the matrix $\mathbf{M} = \mathbf{L}^\dagger \mathbf{T}^{[K,1]}$ with $\mathbf{L}$ the label and $\mathbf{T}^{[K,1]}$ the top tensor (Fig. 1 (d)). This is because the all the tensors in the TTN are orthogonal. Note that $\mathbf{M}$ has four indexes, of which each represents the effective space renormalized from one quarter of the vectorized image. Thus, the bipartition of the entanglement determines how the four indexes of $\mathbf{M}$ are grouped into two bigger indexes before calculating the SVD. We compute two kinds of entanglement entropy by cutting the system in the middle along the x or y direction. Our results suggest that the images of "0" and "4" are the easiest and hardest, respectively, to predict one part of the image by knowing the other part.

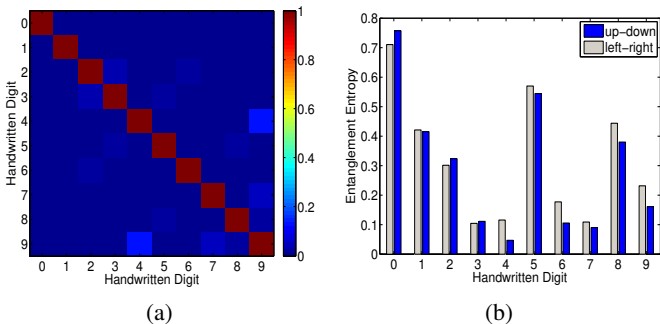

(a)             (b)

Figure 4: (a) Fidelity $\mathcal{F}_{p'p}$ between each two handwritten digits, which ranges from $-0.0032$ to $1$. The diagonal terms $\mathcal{F}_{pp} = 1$ because the quantum states are normalized; (b) Entanglement entropy corresponding to each handwritten digit entropy.

# 6 CONCLUSION AND OUTLOOK

We continued the forays into using tensor networks for machine learning, focusing on hierarchical, two-dimensional tree tensor networks that we found a natural fit for image recognition problems. This proved a scalable approach that had a high precision, and we can conclude the following observations:

- The limitation of representation power (learnability) of the TTNs model strongly depends on the input bond (physical indexes). And, the virtual bond (geometrical indexes) determine how well the TTNs approximate this limitation.
- A hierarchical tensor network exhibits the same increase level of abstraction as a deep convolutional neural network or a deep belief network.
- Fidelity can give us an insight how difficult it is to tell two classes apart.
- Entanglement entropy has potential to characterize the difficulty of representing a class of problems.

In future work, we plan to use fidelity-based training in an unsupervised setting and applying the trained TTN to recover damaged or compressed images and using entanglement entropy to characterize the accuracy.

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
