# OpenReview forum: "Machine Learning by Two-Dimensional Hierarchical Tensor Networks: A Quantum Information Theoretic Perspective on Deep Architectures"
_ICLR.cc/2018/Conference — Reject_

### Official Review · AnonReviewer3 · 2017-11-20
**Authors of this paper derived an efficient quantum-inspired learning algorithm based on a hierarchical representation that is known as tree tensor network, which is inspired by the multipartite entanglement renormalization ansatz approach where the tensors in the TN are kept to be unitary during training.**

**Rating:** 6
**Confidence:** 3

**Review:**

Authors of this paper derived an efficient quantum-inspired learning algorithm based on a hierarchical representation that is known as tree tensor network, which is inspired by the multipartite entanglement renormalization ansatz approach where the tensors in the TN are kept to be unitary during training. Some observations are: The limitation of learnability of TTN strongly depends on the physical indexes and the geometrical indexes determine how well the TTNs approximate the limit; TTNs exhibit same increase level of abstractions as CNN or DBN; Fidelity and entanglement entropy can be considered as some measurements of the network.

Authors introduced the two-dimensional hierarchical tensor networks for solving image recognition problems, which suits more the 2-D nature of images. In section 2, authors stated that the choice of feature function is arbitrary, and a specific feature map was introduced in Section 4. However, it is not straightforward to connect (10) to (1) or (2). It is better to clarify this connection because some important parameters such as the virtual bond and input bond are related to the complexity of the proposed algorithm as well as the limitation of learnability. For example, the scaling of the complexity O(dN_T(b_v^5 + b_i^4)) is not easy to understand. Is it related to specific feature map? How about the complexity of eigen-decomposition for one tensor at each iterates. And also, whether the tricks used to accelerate the computations will affect the convergence of the algorithm? More details on these problems are required for readers’ better understanding.

From Fig 2, it is difficult to see the relationship between learnability and parameters such input bond and virtual bond because it seems there are no clear trends in the Fig 2(a) and (b) to make any conclusion. It is better to clarify these relationships with either clear explanation or better examples.

From Fig 3, authors claimed that TN obtained the same levels of abstractions as in deep learning. However, from Fig 3 only, it is hard to make this conclusion. First, there are not too many differences from Fig 3(a) to Fig 3(e).  Second, there is no visualization result reported from deep learning on the same data for comparison. Hence, it is not convincing to draw this conclusion only from Fig 3.

In Section 4.2, what strategy is used to obtain these parameters in Table 1?

In Section 5, it is interesting to see more experiments in terms of fidelity and entanglement entropy.

---

> ### Author Response · Authors · 2017-12-31
> **The error about feature map was fixed, and we explained the scaling of complexity, the tricks and the strategy we used.**
>
> Thanks for your note. The error about feature map was fixed. Yes, the feature map we used is not arbitrary. In our case, it should be normalized.
> The scaling of complexity is not related to specific feature map. We used nested loop to update each tensor and the tensor contraction dominate main part of computation. For instance, for the tensors located in 1st layer, the complexity of contraction is dominated by input bond b_i, i.e. O(M*N_T*b_i^4*b_v), and for the tensors located in other layers, the complexity of contraction is dominated by virtual bond b_v, i.e. O(M*N_T*b_v^5). Therefore, the complexity of contracting the whole tensor network is composed by these two parts, i.e. O(M*N_T*(b_v^5 + b_i^4*b_v)).
> We used singular value decomposition (SVD) to update each tensor. Considering the SVD on a matrix with M*N (M>>N), the overall cost is O(mn^2). For instance, it could be found in the following book: Trefethen, Lloyd N.; Bau III, David (1997). Numerical linear algebra. Philadelphia: Society for Industrial and Applied Mathematics. ISBN 978-0-89871-361-9.
> The tricks we employed do not affect the computing results and the convergence of the algorithm. We just restore all temporary variables in each iteration, so it avoids double counting.
> In Fig2(a) and Fig2(b), we show the relationship between training accuracy and parameters such input bond and virtual bond. Furthermore, in our classification task, the training accuracy indicates the learnability of the classifier we trained. In other words, the higher training accuracy we have, the better the classifier approximates the classification boundary. So in this case, we say the classifier learn the classification boundary. In our experiments, we fixed the input bond to four different values, i.e. 2, 3, 4 and 5. And we found the significant increase of training accuracy with the rise of virtual bond. And, the input bond determines the upper limits of training accuracy we could have, that is, the upper limits of the learnability of the model.
> The claim is more careful: “We observe the same pattern as in deep learning, having a clear separation in the highest level of abstraction". Naturally, we hoped that the separation would become gradually clearer as in the layers of a CNN, whereas, just as the referee points out, the separation only becomes apparent in the last layer. This is why we were careful with the claim. We did not include a similar series of images for a CNN since this is so standard in the analysis of deep networks. If the referee believes the inclusion of this images would improve the manuscript, we have them ready.
> According to the representation power of tensor network, the larger bond value we used, the better training performance we could have. However, as the same as other existing deep learning models such as deep neural networks, we also have the problem i.e. overfitting. So we start from a small bond value i.e. 2 for each classifier and observed the testing results. Once we have an acceptable result (>92\%), we stop try to use larger value to avoid overfitting.

---

### Official Review · AnonReviewer1 · 2017-11-27
**Novel application of tensor networks**

**Rating:** 4
**Confidence:** 3

**Review:**

Full disclosure: the authors' submission is not anonymous. They included a github link at the bottom of page 6 and I am aware of the name of the author and coauthors (and have previously read their work and am a fan of it). Thus, this review is not double blind. I notified the area chair last week and we agreed that I submit this review.

---

This is an interesting application of tensor networks to machine learning. The work proposes using a tree tensor network for image classification. Each image is first mapped into a higher-dimensional space. Then the input features are contracted with the tensors of the tensor network. The maximum value of the final layer of the network gives the predicted class. The training algorithm is inspired by the multipartite entanglement renormalization ansatz: it corresponds to updating each tensor in the network by performing a singular value decomposition of the environment tensor (everything in the cost function after removing the current tensor to be updated).

Overall, I think this is an interesting, novel contribution, but it is not accessible to non-physicists right now. The paper could be rewritten to be accessible to non-physicists and would be a highly-valuable interdisciplinary contribution.

* Consider redoing the experiments with a different cost function: least squares is an unnatural cost function to use for classification. Cross entropy would be better.

* discuss the scalability: why did you downsample MNIST from 28x28 pixels to 16x16 pixels? Why is training accuracy not reported on the 10-class model in Table 1? If it is because of a slow implementation, that's fine. But if it is because of the scalability of the method, it would be good to report that. In either case it wouldn't hurt the paper, it is just important to know.

* In section 5, you say "input vectors are still initially arranged ... according to their spatial locations in the image". But don't you change the spatial locations of the image to follow equation (10)? It would be good to add a sentence clarifying this.

---

In its current form, reading the paper requires a physics background.

There are a few things that would make it easier to read for a general machine learning audience:

* connect your method to matrix factorization and tensor decomposition approaches

* include an algorithm box for Strategy-I and Strategy-II

* include an appendix, with a brief review of upward and downward indices which is crucial for understanding your method (few people in machine learning are familiar with Einstein notation)

* relate your interesting ideas about quantum states to existing work in information theory. I am skeptical of the label 'quantum': how do quantum mechanical tools apply to images? What is a 'quantum' many-body state here? There is no intrinsic uncertainty principle at play in image classification. I would guess that the ideas you propose are equivalent to existing work in information theory. That would make it less confusing.

* in general, maybe mention the inspiration of your work from MERA, but avoid using physics language when there are no clear physical systems. This will make your work more understandable and easier to follow. A high-level motivation for MERA from a physics perspective suffices; the rest can be phrased in terms of tensor decompositions.

---

Minor nits:

* replace \citet with \citep everywhere - all citations are malformed

* figure 1 could be clarified - say that see-through gray dots are dimensions, blue squares are tensors, edges are contractions

* all figure x and y labels and legends are too small

* some typos: "which classify an input image by choosing"; "we apply different feature map to each"; small grammar issues in many places

* Figure 4: "up-down" and "left-right" not defined anywhere

---

> ### Author Response · Authors · 2018-01-03
> **We have made many efforts on the manuscript so that it could be easily accessible to non-physicists, and some related bugs were also fixed**
>
> Thanks for the suggestions. In the revised version, we have made many efforts on the manuscript so that it could be easily accessible to non-physicists. And, we have tried other cost functions including cross entropy, optimized by algorithms such as full gradient descent, SGD, mini-batch and Adam. The experiments show that the currently-used MERA algorithm (environment + SVD) provides the best accuracy and efficiency. On the other hand, we still cannot embed the cross-entropy cost function into our MERA-inspired algorithm, that is, in the current framework, the cross entropy cost function is incompatible with MERA. In a future study, we will evaluate various algorithms to train the TTN. Thanks for your suggestion.
> We supplemented the training accuracy in Table 1, which is the mean training accuracy among the 10 classifiers. With regards to the downsampling on the MNIST dataset, indeed, we could build a Tree Tensor Network (TTN) on image with 28*28 pixels, but it would increase the complexity of the code. For 2^n * 2^n pixels (with n an integer), we could easily write the code, where the TTN has n layers.
> Regarding the input vectors, we apologize that we did not specify this clearly enough. The input vectors are indeed arranged according to the spatial locations of the pixels.  The feature map transform one pixel (a scalar) to a normalized vector. After the feature map, each image becomes the product of 2^n * 2^n vectors; each vector is located in the same place as the corresponding pixel. We added several sentence in the manuscript to better specify this. And, we have added an algorithm box to specify the one-against-all strategy. And, we added some sentences to explain the upward, downward indices and related notations in the texts.
> In a quantum many-body system, the interactions between particles create quantum entanglement which is considered as new physical resource, so we bridge quantum many-body theory to machine learning, and we hope this will help us to develop some new tools and concepts to explore machine learning. Based on this idea, we employed tensor network to study the basic machine learning task: image classification.
> Regarding the minor nits, we replaced the \citet with \citep, and updated the legends and labels. The “up-down” and “left-right” in Fig. 4 were also explained.

---

### Official Review · AnonReviewer2 · 2017-11-30
**It is unclear how suitable the proposed physics-inspired architecture is, for solving complex machine learning problems.**

**Rating:** 3
**Confidence:** 2

**Review:**

The paper studies the mapping of a mathematical object representing quantum entanglement to a neural network architecture that can be trained to predict data, e.g. images. A contribution of the paper is to propose a 2D tensor network model for that purpose, which has higher representation power than simple tensor networks used in previous works.

There are several issues with the paper:

First, it is hard to relate the presented model to its machine learning counterpart. e.g. it is not stated clearly what is the underlying function class (are they essentially linear classifiers built on some feature space representation?).

The benchmark study doesn’t bring much evidence about the modeling advantages brought by the proposed method. Separating pairs of CIFAR-10 classes is relatively easy and can be done with reasonable accuracy without much nonlinearity. Similarly, an error rate of 5% on MNIST is already achievable by basic machine learning classifiers.

The concept of bond and bond dimensions, which are central in this paper due to their relation to model complexity, could be better explained.

---

> ### Author Response · Authors · 2017-12-31
> **Our main goal in this work is to understand the representation power of a hierarchical network and introduce rigorous metrics from physics to study the model and the data at various levels of the representation.**
>
> We regret that the reviewer does not agree to the relevance of the study. As it is the case with other tensor network-based learning algorithms that recently cropped up, the objective is not to beat a state-of-the-art CNN, at least not yet. Our main goal in this work is to understand the representation power of a hierarchical network and introduce rigorous metrics from physics to study the model and the data at various levels of the representation.
> The surprising finding is not superior performance, but that it works at all and that we see a correspondence between the higher and higher level abstractions that a CNN provides and the two-dimensional tree tensor network. The latter is mathematically well-understood. We believe that this is highly relevant to machine learning and we expect to see more and more research done in this direction, hence we feel that ICLR is in fact the right outlet for this work.
> We are making extensive changes across the manuscript to make it easier to read for non-physicists; please refer to the comments made to the other referees for model details.

---

### Decision · Program_Chairs · 2018-01-29
**ICLR 2018 Conference Acceptance Decision**

**Decision:**

Reject

**Comment:**

This paper seeks to integrate tensor-based models from physics into machine learning architectures. The two main objections to this paper are first that, despite honest (I assume) efforts from the authors, it remains somewhat hard to understand without substantial background knowledge of physics. Second, that the experiments focus on MNIST and CIFAR image classification tasks, two datasets where linear models perform with high accuracy, and as such are unsuitable for properly evaluating the claims made about the models in this paper. Unfortunately, it does not seem there is sufficient enthusiasm for this paper amongst the reviewers to justify its inclusion in the conference.